# The Gender Productivity Gap in Croatian Science: Women Are Catching up with Males and Becoming Even Better

**DOI:** 10.3390/e22111217

**Published:** 2020-10-26

**Authors:** Dorian Wild, Margareta Jurcic, Boris Podobnik

**Affiliations:** 1Department of Finance, Zagreb School of Economics and Management, 10000 Zagreb, Croatia; dorian.wild7@gmail.com (D.W.); mjurcic2@zsem.hr (M.J.); 2Department of Finance, Luxembourg School of Business, 2453 Luxembourg, Luxembourg; 3Faculty of Civil Engineering, University of Rijeka, 51000 Rijeka, Croatia

**Keywords:** power law, Zipf law, gender productivity gap

## Abstract

How much different genders contribute to citations and whether we see different gender patterns between STEM and non-STEM researchers are questions that have long been studied in academia. Here we analyze the research output in terms of citations collected from the Web of Science of males and females from the largest Croatian university, University of Zagreb. Applying the Mann–Whitney statistical test, for most faculties, we demonstrate no gender difference in research output except for seven faculties, where males are significantly better than females on six faculties. We find that female STEM full professors are significantly more cited than male colleagues, while male non-STEM assistant professors are significantly more cited than their female colleagues. There are ten faculties where females have the larger average citations than their male colleagues and eleven faculties where the most cited researcher is woman. For the most cited researchers, our Zipf plot analyses demonstrate that both genders follow power laws, where the exponent calculated for male researchers is moderately larger than the exponent for females. The exponent for STEM citations is slightly larger than the exponent obtained for non-STEM citations, implying that compared to non-STEM, STEM research output leads to fatter tails and so larger citations inequality than non-STEM.

## 1. Introduction

The gender productivity gap in academia, well known as the ‘productivity puzzle’—that on average men publish more papers than women—still persists across the countries, but the trend is ramping down over time and the gap considerably varies between different subfields of science [1,2]. Many theories explain the gender productivity difference ranging from family responsibilities [3,4] to career absences [5], to mention a few.

In contrast to the OECD finding that there are more female than male undergraduate and graduate students in most countries [6], there are relatively few female than male full professors in many countries. According to the US National Science Foundation, female scientists earn almost half the PhDs in science and engineering (STEM fields) in the US, but comprise only 21% of full science professors and surprisingly only 5% of full professors in engineering [7]. There is persisting gender inequality in hiring [8], female scientists continue to face discrimination in earnings [7], funding [9], patenting [10], prizes and awards [11], and grant applications [12]. Moreover, female scientists publish significantly fewer papers in areas in which research is expensive, and it is much less likely that females are listed as either first or last author on a paper [2]. Holman, Stuart-Fox, and Hauser estimate that men are invited by journals to submit papers at approximately double the rate of women [13]. Analyses accomplished in the Quebec region reported that after women have passed the age of about 38, they receive less funding for research than men, and are at a slight disadvantage in terms of the scientific impact (measured by citations) of their publications [14]. However, a few papers reported that the gender gap is diminishing [1,15], e.g., Xie and Shauman reported that the gender differences in research productivity declined overtime with the female-to-male ratio increasing from about 60 percent in the late 1960s to 75 to 80 percent in the late 1980s and early 1990s [15]. Le Moine shows that the concentration of women among researchers who publish a single article is greater than for men, while their representation among ‘star’ scientists is less [16].

How children and marriage affect men and women is not conclusive. Most studies report the positive effect of marriage on scientific productivity, but Prpic [17] shows that men more than women experience benefitsdue to the presence of a spouse. Fox [3] reports that unmarried men are the least productive of all. Married women, particularly those married for the second or third time, exhibit a higher level of productivity. Stack shows that women with preschool-aged children publish less than other women [18] . For example, the divorce rate among tenured females is 50% higher than that of tenured men [19]. Tenured female scientists are almost three times more likely than male colleagues to be single without children [20]. Obviously, the time, energy, and money devoted to raising children can reduce time devoted to science. Due to thinking that raising kids is more a female responsibility, men with children make them more productive than women with children [17]. Moreover, female postdocs who plan or become parents decide to abandon research careers up to twice as often as men [21], confirming that children and marriages do affect carriers in science, but not equally males and females.

In terms of citations, it was revealed that the smaller number of citations received by females is first because on average, men like to cite their own papers 56% more than women [22]. Second, females generally publish less [1]. However, on average, papers written by females receive more citations than papers written by males [23], implying that women often have a higher impact (citations) per publication [24,25,26]. Duch et al. hypothesized and confirmed that the higher the resource requirements on research, the greater is the difference in the publication rates between females and males in favor of males. However, the gender differences in publication rate and citations are discipline-specific [26].

To stress gender differences across different research areas, for the Italian academia the authors showed that females are more productive than male colleagues in medical sciences, agriculture, veterinary sciences, and earth sciences, while men dominate in industrial and information engineering, chemical sciences, physical sciences, and mathematics and information sciences [27]. Abramo et al. concluded that there is a significant number of scientific fields where women’s performance cannot be considered to be inferior compared to men’s.

## 2. Materials and Methods

To test whether the male and female researchers comprise two distinct subgroups and not a unique, in our paper we apply the Mann–Whitney U test that is intended to measure the difference between two populations, in our analyses the difference in ranks of male and female citations. Generally, when the test is applied in practice, the total citations data (males and females collected together) must be first sorted in ascending order. Combining all citations values in a single array, but keeping information of which sample each observation comes from, the U test first ranks all scientists according to their citations from smallest to largest and then separately sums up all female and male ranks, thus, taking the gender into account. We denote these sums by $R_{1}$ and $R_{2}$, whereby $N_{1}$ and $N_{2}$ we denote the respective sample sizes, in our case representing the numbers of male and female researchers. Since the the test is conceived to incorporate both citations and the sample size which is the number of professors of a particular gender, the test is actually designed to evaluate citations per capita of a given gender. The test statistic quantifying the difference between the rank sums is defined as:
(1)$$U_{1}=N_{1}N_{2}+\frac{N_{1}\left(N_{1}+1\right)}{2}-R_{1}.$$
or
(2)$$U_{2}=N_{1}N_{2}+\frac{N_{2}\left(N_{2}+1\right)}{2}-R_{2}.$$
When the two-tail test is applied, the two-tail test statistic *U* is taken to be the smaller of $U_{1}$ and $U_{2}$.

The distribution of U test is symmetrical where a mean and variance equal
(3)$$\mu_{U}=\frac{N_{1}N_{2}}{2},$$
(4)$$\sigma_{U}=\frac{N_{1}N_{2}\left(N_{1}+N_{2}+1\right)}{12}.$$
If $N_{1}$ and $N_{2}$ are at least equal to 8, the distribution of *U* is approximately Gaussian so that
(5)$$z=\left(U-\mu_{U}\right)/\sigma_{U}$$
is standardized Gaussian distributed with mean zero and variance 1.

Here, the null hypothesis is that the distributions of citations per capita of the two subgroups (men and women) are the same. The null hypothesis is rejected for values of the test statistic falling into either tail of its sampling distribution. If a one-tailed test is performed, the alternative hypothesis suggests that the variable of one group is larger than the other group.

## 3. Results

### 3.1. Empirical Evidence

Here, we analyze the publishing career of 3331 active scientists from their publication record in theWeb of Science (WoS) database who have the affiliation of University of Zagreb, which is the largest Croatian university. As a proxy for research excellence, we take the the number of citations according to WoS. With the criterion that only affiliation matters, we exclude the total number of citations the researchers may have collected being on other institutions. The data had been collected between 5th and 10th November 2019. This choice may serve as a proxy for the real contribution of scientists in their university rankings. Due to the importance of physics, biology, and chemistry in science in general, we analyze these subfields of natural sciences individually even though they all belong to a single institution, the Faculty of Natural Sciences. For each scientist we identify the gender, the number of publications, citations, and whether he/she belongs to a STEM field (science, technology, engineering and mathematics) or not. We put particular focus on STEM field (science, technology, engineering, and mathematics) or not. We put particular focus on STEM fields because the recent studies indicate the prominence of STEM for the country’s growth. Specifically, the World Economic Forum [28] and National Academies [29,30] studies indicate that STEM fields are key in economic development.

Among the researchers, we first find 1837 male and 1485 female scientists among the full, associate, and assistant professors. However the gender contribution in different subgroups clearly reveal that academia is becoming more open to females. Specifically, while male full professors substantially outnumber female colleagues (≈50%), i.e., 778 vs. 518, the gender number gap is not only diminishing at the level of associate professor, where there are 415 males and 343 females (≈20% more males), but even brings more females than males at the lowest professor rank, where there are 587 male assistant professors and 613 female assistant professors.

Our focus on bibliometric data limits our analysis to only publishing careers that are easier to analyze than teaching careers. Nevertheless, our efforts constitute an attempt to quantify gender inequality in STEM/non-STEM publications and citations at the largest Croatian university in a former socialist country.

The total numbers of male and female scientists hide an underlying disciplinary differences, as the fraction of women, is as low as 9% in physics, 30% mathematics, and 16% computer science (Faculty of Electrical Engineering and Computing), but female scientists constitute the majority in biology and chemistry with 65% and 56%, respectively (see Table 1). The Faculty of Natural Sciences comprising all-natural sciences, physics, biology, chemistry, including mathematics have 49.49% of male professors.

Specifically, in the US, male scientists are the majority in all STEM fields, in Croatia at the largest university we find that females scientists outnumber their male colleagues in biology and chemistry and this gender equality seems to be a general trend in former socialist countries. For example, Huang, Gates, Sinatra, and Barabasi [31] reports recently a finding that the the proportion of female scientists worldwide can be as low as 28% in Germany and reaches almost 50% in Russia. In support that socialism was helpful in reaching the gender equality in academia, note that according to Shen, in Lithuania, female PhDs recipients in science make 63% of all recipients [7]. Similarly, as on the West, female scientists at the Croatian the largest university has a much larger proportion in social than natural sciences, whereas in some disciplines in social sciences females constitute even the majority in contrast to the US where females’ fraction in social sciences are larger than in STEM fields, but males still substantially dominate. For example, while in the US, females reach 33% in psychology at University of Zagreb female psychologists constitute 71%. Similarly, an interesting result is that at the Faculty of economics females constitute the majority, i.e., 52% (see Table 1).

### 3.2. Empirical Evidence: Males vs. Females at Faculty Level

For each faculty, we apply the Mann–Whitney U test and for most of them, we accept the null hypothesis, thus confirming that at a 5% confidence level there is no gender difference in citations. However, the gender difference in research productivity measured through citations per capita is confirmed for the following faculties (see Table 1): Faculty of Medicine, Faculty of Transport and Traffic Sciences, Faculty of Geodesy, Faculty of Teacher Education, and Faculty of Science-Geology, and Faculty of Law. With a single exception of Faculty of Law, male researchers on these faculties are significantly better than the female. In agreement with Duch et al. our results also reveal that the gender differences in citations are discipline-specific [26].

### 3.3. Empirical Evidence: Males vs. Females at University Level at Different Professor’s Rank

Although women have made significant progress in catching up males concerning the gender productivity gap, on average, female scientists around the world continue to face discrimination. However, a situation on gender equality today is supposed to be much better than a few decades ago. Therefore, next, we test the hypothesis that the gender productivity gap in Croatian academy is diminishing over time. Gender inequality can be nicely captured analyzing the productivity and impact differences quantified by citations between the genders. As a good dynamic proxy for females’ catching up males in the following analyses, we make gender comparison separately for the assistant, associate, and full professors. In Table 2 for the top 500 researchers we find that while on average, male full professors publish 23 papers during their active career, female full professors publish almost the same number of publications, 20, resulting in a small gender gap in total productivity. At a rank of associate professor, we find no difference in total productivity between the genders, but as an interesting result, we find that females have on average more citations than the male colleagues (see Table 3), an outcome to our knowledge not reported for any other country. Again, a difference between the genders appears between the male and female assistant professors, but this is most likely since raising a family is still more female than male responsibility.

### 3.4. Empirical Evidence: STEM Males vs. Females at University Level at Different Professor’s Rank

Hence, as humans’ societies are developing the difference in science performance between males and females is diminishing. However, it is not clear whether females are approaching males in their scientific results equally in STEM and non-STEM fields. Holman, Stuart-Fox and Hauser estimate that the gender gap appears likely to persist for generations, particularly in surgery, computer science, physics, and maths [13] .

Making comparison first for STEM fields, we obtain that male colleague exhibit significantly better results than their female colleagues only for a rank of full professors, while the gender difference is fading away for the associate and assistant professors. Please note that in the Mann–Whitney test, the larger the difference between the two populations, the larger the difference between the *U* values. To test whether STEM female and male researchers comprise two different subgroups, we apply the Mann–Whitney U test that quantifies the difference between two populations based on the difference between the ranks of their citations. Here the null hypothesis is that the distributions of the two subgroups are the same. We find that for full professors the test statistics produce *z* score =−3.79 (Table 4), rejecting the null hypothesis and confirming that at a 5% confidence level female STEM full professors are significantly more cited than their male colleagues. However, the gender gap we do not find for the other two ranks of a professorship, implying that STEM fields presently are more popular among females than a few decades ago.

### 3.5. Empirical Evidence: Non-STEM Males vs. Females at University Level at Different Professor’s Rank

The similar comparison between males and females we perform in non-STEM fields. We find no gender differences at the levels of full and associate professorship. However, we obtain that for the youngest level of the professorship, males exhibit significantly better results than their female colleagues. From Table 5 we find that for assistant professors the *z* score =2.54, rejecting the null hypothesis that the distributions of the two subgroups for males and female are the same and confirming that at a 5% confidence level non-STEM male assistant professors are significantly more cited than their female colleagues. The result is not surprising because female assistant professors are at the age when as young women they spend a considerable amount of time with their kids. This result is in agreement with the conclusions by Stack, who found that women with preschool-aged children publish less than others [18].

### 3.6. Empirical Evidence: Pareto Inequality, STEM vs. Non-STEM

Since the work of Pareto on the application of power law to income inequality, we know that the tail part of an income distribution follows a power law [32]. Since that discovery, a huge literature has been published to demonstrating how distributions in economics and generally social science follow power-law tails and contribute to inequality [33,34,35,36,37]. As Stiglitz realized [36], the small fraction of superstar firms that dominate entire sectors of the economy (e.g., Amazon, Apple, Google, Microsoft) are the main drivers of wealth disparity. Indeed, it seems they are outliers, driven by STEM. Here we can hypothesize that STEM fields generate the fatter tails than non-STEM fields even in academia, when citations are considered. Motivated by Redner’s finding that citations follow power law [38], here we apply the power-law formalism to test the research inequality between male and female scientists at the University of Zagreb, with particular focus on the difference between STEM and non-STEM fields.

Figure 1 shows the log-log Zip plots of citations versus rank [39,40], which is an alternative representation of the Pareto distribution [39,40]. When a probability distribution is asymptotically represented by a Pareto (power-law) distribution with exponent 1+α, then a Zipf plot of size *s* versus rank *R* asymptotically follows a power law with exponent ξ relating the Pareto exponent α as [41]
(6)ξ=1/α.
Generally, the smaller the α value, the fatter the tail of the power-law distribution.

Here we apply the Gabaix-Ibragimov R−1/2 method of fitting Zipf plots [40]. For year 2020, we perform a Zipf plot to both female and male researchers, no matter whether they are in STEM and non-STEM fields, and find the Zipf exponent for females ξ=0.62±0.01 corresponding to α=1.61, whereas for males ξ=0.67±0.01 corresponding to α=1.49. Here we note that both α values are within a range α∈(0,2) characteristic for Levy distributions [41], which are characterized by infinite variance for which is known that the famous *Central Limit Theorem* does not hold.

Next we extend our analysis on research output, measured by citations, to separate STEM and non-STEM fields. Figure 2 shows a small difference between the slopes of the Zipf plot of STEM citations and non-STEM citations, $\xi_{S}=0.66>\xi_{nS}=0.61$, thus STEM researchers generate fatter tails than non-STEM researchers (see Equation (Equation 6)).

Statistics of citations for all male and female researchers we show in Table 6, while in Table 7 and Table 8 we show statistics for STEM and non-STEM, respectively. In STEM, female associate professors have larger mean and kurtosis than their male colleagues. Kurtosis is generally larger for STEM than for non-STEM fields.

The Pareto principle has been found in large number of datasets. This principle, accomplished for citations, states that roughly 80% of consequences come from 20% of the causes [32,42]. However this principle does not hold for any power law, but for α=1.16 for which 80% of effects come from 20% of causes.

From Table 9 for male STEM researchers we find that 20% of males bring 80% of citations in well agreement with Pareto principle. However, 27.2% of STEM females bring 80% citations. In case of non-STEM researchers we find substantial difference from the Pareto principle.

Shannon’s entropy of information is a foundational concept in information theory quantifying the amount of information embedded in the variable or the amount of storage expressed by the number of bits required to store the variable [43]. The larger the entropy, the larger the amount of surprise comprised in the data. When all values of discrete variable (*R* in total) are equally probable, each probability equals 1/R and then it is easy to show that the Shannon entropy equals ln(R). If the probability of a particular variable value approaches 1, the Shannon entropy tends to zero.

Here we apply Shannon’s entropy for a randomly chosen citation where it can be assigned to either males or females. In Table 10 the last column reports the entropy values which are close to the entropy (equal to 1) which one would obtain for a regular coin with two equally likely outcomes for which there is no way to predict the outcome of the coin toss. The entropy equal to 0 would represent a case when one of the probabilities equals 1 and any foredooming outcome can be predicted perfectly. We demonstrate no gender difference except small differences at the level of full professors, both STEM and non-STEM, and at the level of STEM Assistant Professors.

## 4. Discussion and Conclusions

At the largest Croatian University, applying the Mann–Whitney test, we demonstrate no gender difference in number of citations except for seven faculties, where males are significantly better than females on six faculties. We report that female STEM full professors are significantly more cited than their male colleagues, while male non-STEM assistant professors are significantly more cited than their female colleagues. In Table 11 as a comparison between male and female researchers, we report the average citations across different University members. In contrast to the US and many Western nations, at the largest Croatian University there are several faculties where the most cited researcher is woman and these faculties are the following, both STEM and non-STEM: Faculty of Chemical Engineering and Technology, Faculty of Economics and Business, Faculty of Food Technology and Biotechnology, Faculty of Geotechnical Engineering, Faculty of Graphic Arts, Faculty of Kinesiology, Faculty of Organization and Informatics, Faculty of Political science, Faculty of Science-Biology, Faculty of Science-Geography, and Faculty of Teacher Education. Thus, at individual level, at some particular faculties and scientific fields, the most cited female is better off than the most cited male. Moreover, at collective level, there are ten faculties where females have the larger average citations than their male colleagues: Faculty of Chemical Engineering, Faculty of Economics, Faculty of Food Technology, Faculty of Forestry, Faculty of Geotechnical Engineering, Faculty of Graphical Arts, Faculty of Organization and Informatics, Faculty of Political science, Faculty of Science-Physics, and Faculty of Dental Medicine.

Wealthy countries such as Japan, Germany, and Switzerland has fewer women authors than poorer ones [13]. Besides this asymmetry, there is a significant difference in gender productivity between the US and the poorer countries. However, would it not be reasonable to assume that the more developed a country, the smaller the gender differences in their performance? Partially the gender difference in the US academia appears because in the US, in contrast to the EU, the large majority of universities are private and the private educational system recognizes only quality and does not care much about gender or even racial equality. Comparing the US and developing countries, due to globalization where the US is the attractor of best students all over the world, the level of competition in the US academia is much higher than in any other country. Therefore, females to compete with males in the US academia must compete not only with the best US males, but with the most brilliant world candidates. Since raising a family is even in the US still considered to be more female than male responsibility, being a female makes their position much harder. Clearly, if a female is a genius like Nobel Prize Laureate Marie Curie, she can achieve excellent academic results in parallel with being an excellent mother and even raise her daughter to become another Nobel Prize Laureate. However, there are not so many geniuses of that kind. In large educational systems such as the US, fluctuations are substantially smaller than in small countries such as Croatia, and so events much different than expectations are more likely.

Briefly, we expect that in a limit where children are equally educated regardless of gender and where family obligations are equally distributed among both parents, it is meaningful that men and women should perform equal results, not only in science. The closer a society to this limit, the larger the similarity in their performance. It is a challenge to test this obvious hypothesis. To this end, in academia, the gender research productivity gap will not diminish without substantial reforms in education, mentoring, and academic publishing.

For a country to be competitive in both science and business, in agreement with Cobb-Douglas production function extended for human capital, maximization of its intellectual capital particularly its females’ part is a top country’s priority. In the short run, policymakers should launch new programs required to stimulate international collaboration for female researchers, because international collaboration is one of the pillars for excellence in science. We suggest that the best female researchers should generally receive more grants and funding than their best male colleagues as compensation for spending more time in raising a family. The pay gap in science should become an illegal practice. In the grant review process, gender equality among reviewers should be always respected, and if not, the call should be canceled. We even suggest that kindergartens should be open at least within large Universities and institutes. In long run, we need to rapidly boost the interest in science and particularly STEM disciplines among the girls.

## Figures and Tables

**Figure 1 entropy-22-01217-f001:**
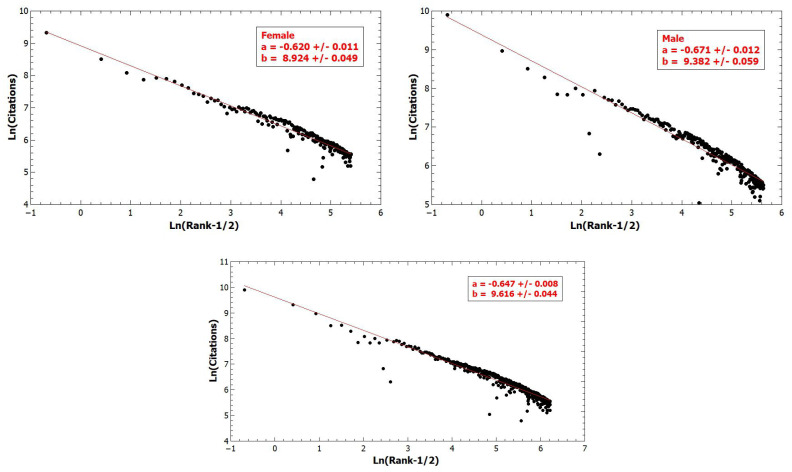
No substantial difference in research output measured by citations between males and females. However, the Zipf plot of citations for males exhibit a slightly fatter tail than the females’.

**Figure 2 entropy-22-01217-f002:**
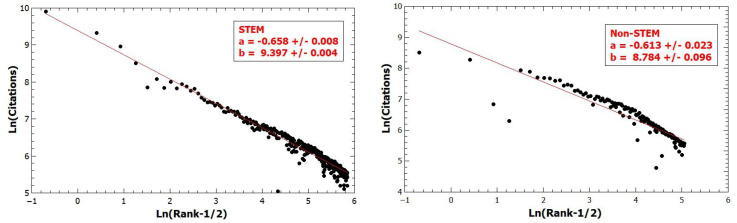
STEM researchers contribute to a slightly fatter tails thansdcc non-STEM researchers.

**Table 1 entropy-22-01217-t001:** Men vs. Women—citation comparison across different University constituents. Shown are *z*-scores of Equation (Equation 5) and corresponding *p*-values for both one- and two-tailed statistics where in brackets we indicate if there is dominance of males or females.

	% of Females	*z*-Value	*p*-Value
Catholic Fac of Theology	34.4	0.022	0.982
Fac Agriculture	48.4	−1.95	0.05 (0.025) (M)
Fac. Architecture	31.3	0.872	0.383
Fac Chem Eng & Tech	56.3	0.255	0.799
Fac Civil Eng	44.2	0.344	0.730
Fac Econ & Business	51.6	0.128	0.896
Fac Education & Rehabilitation Sci	87.3	1.069	0.285
Fac Electri Eng & Comp	16.2	0.359	0.720
Fac Food Tech & Biotech	69.4	0.202	0.840
Fac Forestry	18.5	0.065	0.948
Fac Geodesy	14.3	2.268	0.023 (0.11) (M)
Fac Geotechnical Engineering	26.1	1.330	0.182
Fac Graphic Arts	63.3	0.367	0.714
Fac Humanities & Social Sciences	51.9	0.297	0.766
Fac Kinesiology	44.3	0.951	0.341
Fac Law	54.1	−2.259	0.023 (0.12) (F)
Fac Mech Eng & Naval Arch	14.2	0.567	0.571
Fac Metallurgy	47.8	−1.848	0.064
Mining, Geology & Petroleum Eng	36.3	1.874	0.061
Fac Organization & Informatics	41.1	0.226	0.821
Fac of Pharmacy & Biochemistry	81.3	0.791	0.429
Fac Political Science	38.7	1.376	0.169
Fac Science—Physics	9.1		
Fac Science—Biology	65.4	0.987	0.322
Fac Science—Chemistry	56.1	0.631	0.529
Fac Science—Geography	38.5	0.847	0.397
Fac Science—Geology	54.5	2.078	0.038 (0.19) (M)
Fac Science—Geophysics	50	1.123	0.317
Fac Science—Mathematics	29.6	0.786	0.432
Fac Teacher Education	63.5	2.108	0.035 (0.17) (M)
Fac Textile Technology	60.9	1.150	0.250
Fac Transport & Traffic Sciences	25.4	2.321	0.020 (0.01) (M)
Fac Veterinary Medicine	43.5	0.822	0.411
Fac Dental Medicine	53.6	0.910	0.362
Fac Medicine	46	−2.711	0.006 (0.003) (M)

**Table 2 entropy-22-01217-t002:** Men vs. Women—Publications comparison. Both STEM and non-STEM.

	Average No. Pubs (M)	Average No. Pubs (F)
Assistant Prof	12	8.9
Associate Prof	15	14.7
Full Prof	23.4	19.8

**Table 3 entropy-22-01217-t003:** Men vs. Women—Citation comparison. Both STEM and non-STEM.

	Average Citations (M)	Average Citations (F)
Assistant Prof	97.7	64.9
Associate Prof	127	144.7
Full Prof	258.2	207

**Table 4 entropy-22-01217-t004:** STEM: Men vs. Women—citation comparison.

	No. (M)	No. (F)	U (M)	U (F)	*z* Score	*p*-Value
Assistant Prof	331	248	40,284	41,804	0.38	0.704
Associate Prof	214	174	19,980	17,256	−1.24	0.215
Full Prof	511	266	79,219	56,706	−3.79	0.00

**Table 5 entropy-22-01217-t005:** Non-STEM: Men vs. Women—citation comparison.

	No. (M)	No. (F)	U (M)	U (F)	*z* Score	*p*-Value
Assistant Prof	256	365	41,286	52,156	2.54	0.01
Associate Prof	201	169	16,739	17,230	0.24	0.80
Full Prof	267	252	33,256	34,027	0.22	0.819

**Table 6 entropy-22-01217-t006:** Men vs. Women—Statistics. Both STEM and non-STEM.

	Mean	Median	Skewness	Kurtosis
All	161.6	31	21.5	687.8
All male	180.5	35	19.4	508.9
All female	138.2	26	6.7	117.9
Assistant Prof (M)	97.7	20	21.4	494.9
Assistant Prof (F)	64.9	10	7.6	89.5
Associate Prof (M)	126.9	33	4.3	24.2
Associate Prof (F)	144.7	35	9.3	11.9
Full Prof (M)	258.2	59	16.3	341.9
Full Prof (F)	207.0	61.5	4.04	124.0

**Table 7 entropy-22-01217-t007:** Men vs. Women—Statistics. STEM.

	Mean	Median	Skewness	Kurtosis
All male	208.7	51	17.9	395.9
All female	197.9	68.5	6.3	160.5
Assistant Prof (M)	122.7	30	16.8	297.6
Assistant Prof (F)	94.9	30	7.5	78.1
Associate Prof (M)	141.7	44.5	3.7	116.9
Associate Prof (F)	181.8	70	9.2	101.7
Full Prof (M)	281.4	73	15.4	280.6
Full Prof (F)	278.7	142	3.5	120.8

**Table 8 entropy-22-01217-t008:** Men vs. Women—Statistics. non-STEM.

	Mean	Median	Skewness	Kurtosis
All male	138.6	9	6.8	62.8
All female	85.4	6	6.2	51.8
Assistant Prof (M)	65.4	6	4.3	23.9
Assistant Prof (F)	44.5	3	5.4	35.9
Associate Prof (M)	111.3	19	4.9	28.6
Associate Prof (F)	106.5	9	4.5	25.8
Full Prof (M)	213.7	11	5.2	37.0
Full Prof (F)	131.2	12.5	5.3	34.6

**Table 9 entropy-22-01217-t009:** Men vs. Women—Pareto principle. Both STEM and non-STEM.

	STEM (M)	STEM (F)	Non-STEM (M)	Non-STEM (F)
	20.5	27.2	14.13	15.2

**Table 10 entropy-22-01217-t010:** Men vs. Women—Shannon’s Entropy. Both STEM and non-STEM.

	Probability (M)	Prob (F)	Shannon’s Entropy
STEM Assist Prof	0.6331	0.3668	0.9482
STEM Associate Prof	0.4892	0.5107	0.9996
STEM Full Prof	0.6594	0.3401	0.9249
Non-STEM Assist Prof	0.5076	0.4923	0.9998
Non-STEM Associate Prof	0.5540	0.4459	0.9915
Non-STEM Full Prof	0.6330	0.3669	0.9482
Assistant Prof	0.5904	0.4095	0.9762
Associate Prof	0.5148	0.4851	0.9993
Full Prof	0.6520	0.3479	0.9322

**Table 11 entropy-22-01217-t011:** Men vs. Women—average citation across different University constituents.

	Average Citations for Males	Females
Catholic Fac of Theology	18	1.0
Fac Agriculture	146.1	89.8
Fac. Architecture	4.5	3.3
Fac Chem Eng & Tech	411.7	528.9
Fac Civil Eng	46.2	31.4
Fac Econ & Business	13.6	19.2
Fac Education & Rehabilitation Sci	23.1	7.7
Fac Electri Eng & Comp	129.9	111.0
Fac Food Tech & Biotech	259.0	320.8
Fac Forestry	75.4	77.4
Fac Geodesy	23.9	3.6
Fac Geotechnical Engineering	16.8	82.5
Fac Graphic Arts	23.6	49.9
Fac Humanities & Social Sciences	24.4	11.7
Fac Kinesiology	186.6	160.6
Fac Law	18.5	11.8
Fac Mech Eng & Naval Arch	116.3	91.2
Fac Metallurgy	129.1	33.5
Mining, Geology & Petroleum Eng	89.5	44.4
Fac Organization & Informatics	14.2	21.4
Fac of Pharmacy & Biochemistry	640.3	387.8
Fac Political Science	6.2	13.3
Fac Science—Physics	1931	2365
Fac Science—Biology	372.8	361.6
Fac Science—Chemistry	488.2	390.4
Fac Science—Geography	22.3	21.4
Fac Science—Geology	206.1	83.3
Fac Science—Geophysics	611.7	215.7
Fac Science—Mathematics	167.4	87.6
Fac Teacher Education	16.4	7.6
Fac Textile Technology	254.7	74.9
Fac Transport & Traffic Sciences	17.5	4.4
Fac Veterinary Medicine	203.0	149.2
Fac Dental Medicine	140.2	150.7
Fac Medicine	442.3	285.5

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
