# Peer review of "The Gender Productivity Gap in Croatian Science: Women Are Catching up with Males and Becoming Even Better"

_entropy, 2020, doi:10.3390/e22111217_

Round 1

Reviewer 1 Report

I read a paper titled The Gender Productivity Gap in Croatian Science: Women are Catching up with Males and Becoming even Better.

The authors consider the research gap about gender productivity at the University of Zagreb. They specify a set of hypotheses and find these confirmed. The implication of this paper is clear, however, I have a few minor observations

 Literature: please provide a comprehensive table  of ten most cited papers  which are dealing with the same problem and their findings

 add limitation section, practical implication section and policy recommendations. 

On the technical side, I would urge the authors also to provide measures of fit ( number of authors per paper, etc). Additionally, please in appendix add the list of up to five names per faculty and field according to your findings. 

Thank you for the authors to give me an opportunity to read an interesting paper.  

Author Response

 We are grateful to all valuable comments. 

Reviewer 2 Report

The paper is devoted to data on the scientific activity of female and male professors in the Zagreb University.
The data are collected in November 2019. Some statistical hypotheses are analyzed there; all about the differences
between both genders.

The paper consists of four parts. First (about one page) is a report of the state of art, and it is fairly reasonable.
Second (till page 5, a part of left column) is the description of data and some inferences, and it seems credible.
Third (a 'toy' model, a half of page) provides some parameterization, differently for females and males. Last part
is devoted to conclusions.

My objections are against last two parts. The model proposed is a set of parameters, entirely unrelated to the collected data.
Also, the model remains completely unexplored. Equations are written without any purpose, and the Authors
did not try to evaluate any parameter. Further, the discussion is a loose set of statements, also without contact with the
data and even with the model. In my opinion, this kind of presentation does not suit for publication.

However, I can imagine that the data collected can be publishable in some journal, if the model is either successfully applied
(what is more difficult) or entirely removed (what is simpler) and if the conclusions are related to the Zagreb data.

Minor points:

page 2, left column, the Authors write: "Even though ...equally biased". I disagree: the bias depends on gender differently
for STEM (prevailing in WoS) and for other areas. The sentence is not justified.

page 4, left column, the Authors write: "This result contradicts ..." Actually, there is no contradiction between "publish
less" and "are more cited".

construction of some sentences unfinished:

page 2, left column, last sentence of the first paragraph: "is at least" unnecessary

page 4, left column: "However, the gender gap ...decades ago."

page 5: "According to Boschma... (sentence completely unclear; too long?)

misprints:

page 2: wether > whether

Table 1: Geotehnical > Geotechnical

page 5, right column: re reasonable > reasonable (and why the question mark just before this?)

Also, please evade writing "fascinating" about your own results (page 4, left column)

Author Response

 Based on you valuable comments we decide to delete the entire model part. 

We corrected  typos. 

Reviewer 3 Report

The article deals with gender inequality as seen from data obtained from the University of Zagreb and the Web of Science, comparing it to the ones of other countries, mainly the USA. The subject is worth of investigation and, as the authors cite, it deals with a former socialist country, and it may present differences due to many years under the rule of this regime, that may have many negative aspects, according to perspective, but that was particularly keen on gender equality.

In what follows, I enumerate some of my major concerns.

1 – The foremost major concern is that I don’t see how the article relates with the line of publication of Entropy. There is some small part of it dealing with Zipf’s law, but no mention at all to something that resembles entropy, except for the logarithmic terms.

2 – The second major concern is that, for me, it seems like there are two articles, written mostly by different researchers. The first part is objective, data-driven, and well written. The second part, from “A Toy Productivity Gap Model” to the end of the conclusions, has lots of guesses (educated guesses, but guesses) that are merely based on common assumptions, that may be right, but that are not based in any perceivable data or certified facts.  There are differences in style as well, with the second part containing a good number of typos, what is not present in the first part of the article.

3 – At the end of the conclusions, it is talked about proving an obvious hypothesis. Maybe the word “obvious” should be avoided in a scientific article.

4 – The authors present a model “A Toy Productivity Gap Model” that seems somehow disconnected from the previous part of the article and does not seem to be tested with data.

5 – The authors cite results on the tails of IQ tests, and say it may explain better results of male students in sciences. Maybe this affirmation should be backed by other results or data, since it leads to the idea that better IQ levels lead to better results in sciences but not necessarily in other fields of research.

Concluding, it is of my opinion that more attention should be given to the second part of the article, making it more objective and more in line with the first part. If possible, the connection of the research with the main topics of the journal should also be stressed.

I’ve spotted some typos/mistakes that I cite below. My recommendations are to change, at the authors’ discretion:

  • “to test the far tails” to “to test the fat tails”;
  • “avoidincertain” to “avoiding certain”;
  • “random wall” to “random walk”;
  • “considerable time time” to “considerable more time”;
  • “For that reason in a model” to “For that reason, in a model”;
  • “and the poorer countries?” to “and the poorer countries.” (Is there a citation missing here?);
  • “But wouldn’t re reasonable” to “But wouldn’t it be reasonable”;
  • “is genius as Nobel Prize Laureate Marie Curie she” to “is a genius like Nobel Prize Laureate Marie Curie, she”.

Round 2

Reviewer 2 Report

The paper is much better and in my opinion it can be published almost in the present form. I found only some misprints :
- page 5, right column: it be e reasonable > it be reasonable
- acknowledgements: Ctoatian > Croatian
- references: missing commas in refs 24-26, errors in ref 32, ? > - in ref 38, Humen > Human in ref 39. Please check all references, now the edition is not consistent.

Author Response

 We are grateful to all the valuable comments. We corrected all the mistakes. Thank you very much  

Reviewer 3 Report

Unfortunately, the issue of the adequacy of the article to Entropy has not been resolved. Authors just cite the relevance of the topic, of which I have no doubt.

With respect to the part of the article I deemed different, I don't think removing it makes the article stronger. Some more work put into that part would have been more appropriate. So, I'm sorry, but I cannot recommend this article for publication in Entropy.

Author Response

 Motivated by the Referee suggestions,  we enter  more statistics, apply Shannon's entropy and Pareto principle. All changes are shown in colour in the pdf of the paper. Shown are both Tables and new text. 

  Regarding the model part, we decided not to work on the extension since one of the Referees suggested to remove it. 

  We are grateful to all the comments.